# Are the benefits of prosocial spending and buying time moderated by age, gender, or income?

Iris Lok *, Elizabeth W. Dunn

Department of Psychology, University of British Columbia, Vancouver, Canada

* iris.lok@psych.ubc.ca

## Abstract

In the last two decades, social psychologists have identified several key spending strategies that promote happiness such as making time-saving purchases (buying time) and spending money on others (prosocial spending). Although the emotional benefits of these two spending strategies are well-documented in the current literature, it is unclear whether the effectiveness of these strategies vary depending on individual characteristics. To address this research gap, we surveyed an economically diverse sample of 15,545 Americans about their subjective well-being, spending behavior, personal values and beliefs, as well as demographics including age, gender, and income. Across demographic groups, spending money on others was robustly related to happiness. Spending money on others was also associated with greater happiness regardless of whether participants believed that they would be happier spending money on others. In contrast, the relationship between buying time and happiness was somewhat less reliable. Although gender and personal income did not moderate the relationship between buying time and happiness, the relationship was only marginally significant for men, and non-significant within each income bracket. Our results also indicated that those who valued money over time were significantly happier when they used money to buy time, whereas those who valued time over money reported similar levels of happiness whether or not they bought time. Taken together, the present research shows that the relationship between prosocial spending, buying time, and subjective well-being is largely consistent across the different demographic groups we examined.

## Introduction

In the last two decades, social psychologists have identified several key spending strategies that promote happiness [for a brief review, see 1]. For example, spending money on others (also known as *prosocial spending*) and making time-saving purchases (also known as *buying time*) can lead to higher levels of subjective well-being [SWB; e.g., 1,2]. These two practical spending strategies have received significant attention in the media [e.g., 3–8]. However, it is unclear whether engaging in prosocial spending or buying time should be widely recommended. Indeed, past research has shown that the effect of buying experiences—another well-known

**Competing interests:** The authors have declared that no competing interests exist.

spending strategy for increasing happiness—depends on individual characteristics, such as income [9], personality [10], and social class [11]. Thus, the goal of the current research was to examine whether two key spending strategies—prosocial spending and buying time—are consistently linked to greater SWB across different demographic groups in the United States.

## What spending strategies promote happiness?

**Buying experiences.** The earliest research on the ways in which money can be spent to promote happiness looked at the emotional impact of buying experiences: purchases that are primarily motivated by a desire to acquire a life experience, such as buying a meal at a restaurant or tickets to a major sporting event. Across many correlational and experimental studies, researchers have found a robust relationship between buying experiences and happiness [for a review, see 12]. However, researchers have also identified important boundary conditions for this effect. For example, Van Boven and Gilovich [9] found that the "experiential advantage" was not present among individuals at the lowest end of the income distribution. More recently, Lee, Hall, & Wood [11] found that the emotional benefits of buying experiences was completely absent among individuals from lower social class backgrounds. These findings point to the important possibility that the advice to spend more on certain types of purchases (e.g., experiences) can sometimes be misguided—depending on who is listening.

**Spending money on others.** Looking beyond buying experiences, subsequent studies have shown that people also feel happier when they spend money on others compared to themselves [13–18]. In a recent registered replication report, for example, 730 students were randomly assigned to make a purchase for themselves or someone else [15]. Consistent with early findings, participants who made a purchase for someone else felt happier than those who made a purchase for themselves.

Although the relationship between prosocial spending and well-being is relatively well-established, research on whether this effect is moderated by demographic characteristics is relatively scarce. Specifically, researchers have primarily focused on the moderating role of income. For example, Aknin and colleagues [14] used data from the Gallup World Poll—a representative, large-scale survey that collects responses from more 250,000 people all across the world—to test whether donating to charity in the past month was associated with greater life satisfaction in 136 countries. People who donated in the past month reported being more satisfied with life—in poor and rich countries alike. However, the researchers did not examine whether this relationship between prosocial spending and life satisfaction is moderated by differences in *personal* income.

**Buying time.** More recently, researchers have considered a third strategy for increasing happiness: buying time. In a field experiment, Whillans and colleagues [2] showed that working adults who were assigned to make a time-saving purchase (e.g., taking a cab, ordering takeout) reported being in a better mood compared to those who were simply told to make a material purchase (e.g., buying clothes, books and other items). Correlational studies have also established a consistent link between buying time and life satisfaction in several countries including the US, Canada, and Netherlands. For example, adults who pay for help on disliked tasks (e.g., household chores, shopping) report greater levels of life satisfaction, even after controlling for key covariates such as income, age, and gender [2].

Do demographic characteristics moderate the emotional benefits of buying time? Combining across six different studies with more than 4,000 respondents, Whillans et al. [2] found no evidence that the relationship between buying time and life satisfaction was moderated by income or wealth. This (null) finding suggests that people who spend money on time-saving purchases are more satisfied with life regardless of their income. However, these studies were

not designed to test for interaction effects, and to the best of our knowledge, no other work has examined whether the emotional benefits of buying time are moderated by income. In a similar vein, little is known about whether the benefits of buying time are moderated age or gender [e.g., 19].

## Spending strategies and happiness: Potential moderators

To date, very few studies have identified potential moderators for the relationship between prosocial spending, buying time, and SWB. This is perhaps unsurprising in light of recent research showing that massive sample sizes are required to detect interactions in which a moderator reduces a main effect without reversing it [20]. In a typical 2x2 factorial design, for example, more than a total of 2,500 participants are needed to reliably detect an attenuated interaction with a medium effect size. Addressing this research gap, the current research harnessed a dataset with over 10,000 people to test whether the relationship between different spending strategies and SWB vary with age, gender, and personal income.

Why might demographic characteristics—such as age, gender, and personal income—moderate the relationship between spending strategies and SWB? For more than 40 years, researchers have documented a robust relationship between age and prosocial behavior. For instance, older adults tend to behave in a more prosocial manner compared to their younger counterparts, controlling for covariates such as income and religious involvement [for a review, see 21]. One interpretation of these findings is that engaging in prosocial behavior is more emotionally rewarding for older adults. Thus, we explored whether age moderated the relationship between prosocial spending and SWB.

According to socioemotional selectivity theory, people also become increasingly aware of the limits of their time as they grow older [22]. Thus, making time-saving purchases may be associated with greater SWB among older adults because it frees them from spending their limited time on disliked tasks. At the same time, because older adults are more likely to be retired, they might have more free time—and feel more time affluent—compared to their younger counterparts. As a result, they may reap fewer benefits from buying more time. In the current research, we explore these possibilities by testing if and how age moderates the relationship between buying time and SWB.

The relationship between spending strategies and SWB may also vary with other important demographic characteristics. For example, psychologists have recently been called to pay more attention to the role of gender in research [e.g., 23]. Perhaps because of the large sample sizes needed to detect moderating effects, existing studies have not been designed to identify the dimensions on which the relationship between spending and SWB varies. Thus, we examined the moderating role of gender and personal income in our current research.

Looking beyond demographic variables, we also examined whether prosocial spending and buying time are only associated with greater levels of SWB when people are spending money in ways that fit their personal values and beliefs. Whillans et al. [24] showed that people vary in their resource orientation towards time and money: while some people prioritize having more time than money, others prioritize having more money than time. Likewise, people may vary in their beliefs about what will bring them the most happiness: spending money on themselves versus spending money on others. Because the benefits of prosocial spending and buying time may depend on whether each of these spending strategies are endorsed by participants, we tested (i) whether resource orientation moderates the relationship between buying time and SWB, and (ii) whether lay theories about spending money on others moderates the relationship between prosocial spending and SWB.

### The current research

In a sample of over 15,000 participants, we explored whether the relationship between spending strategies and SWB was moderated by demographic characteristics, and personal values and beliefs. Compared to buying experiences, what moderates the relationship between prosocial spending, buying time, and SWB has received less empirical attention. Thus, we focused on these two strategies in the current research.

## Methods

### Overview

We used data that were collected as a part of an internal research effort by Happy Money, a US financial technology company, in 2019. In line with their privacy policy, Happy Money was permitted to share anonymized data with external parties. Because the data were analyzed anonymously, we did not obtain consent from participants who completed the survey. We also obtained permission to conduct secondary data analyses from our university ethics board (H20-00369). Some of the items in the survey were adapted so that they aligned with the branding of the company. To ensure that the adapted measures were reliable and valid, the company recruited a separate sample of 1000 participants ($M_{age}$ = 39.26 years old, $SD_{age}$ = 11.53; 41.84% Female, 58.16% Male; 76.88% White, 12.51% Black, 5.31% Asian, 3.30% Hispanic/Latino, 2.00% Other) to complete both the original and adapted measures of SWB. We report the correlations between the original and adapted survey items throughout the methods section. All materials, raw data and codebooks are available at https://osf.io/eqadp/.

### Participants

Participants were 15,545 American adults who had previously applied for a loan with Happy Money. The mean age of the sample was 37.40 years old (SD = 12.81). 79.25% identified as female, 20.28% identified as male, and 0.46% identified as neither female nor male. The median income was $46,117 and varied widely in the sample (SD = $26,771). In addition, most of the sample reported living from paycheck to paycheck (88.64%), allowing us to examine the relationship between spending and well-being among people with little discretionary income. No participants were excluded from data analysis.

### Procedures

Participants were invited to complete a survey about money and happiness via e-mail and advertisements on social media. In exchange for their participation, all participants were entered into a raffle to win cash prizes (with top prizes worth up to $2000USD each). Participants completed a series of questions about their SWB, spending behavior, resource orientation towards time and money, beliefs about spending money on others, as well as demographics (see Table 1 for correlation table). In this article, we describe the measures that are most relevant to our current research questions; the full survey administered by the company can be found at https://osf.io/y5wxt/. To make the survey more engaging for participants, the company adapted existing survey instruments by modifying the items and response options included in the survey.

### Measures

**Subjective well-being.** In his seminal work, Ed Diener [25] defined subjective well-being (also referred to as happiness) as a combination of two distinct but related components: a person's *affective* and *cognitive* evaluation of their own life. In line with this widely accepted definition, the company included measures of mood and life satisfaction in their survey. To

**Table 1. Correlation table for examined variables.**

| | 1 | 2 | 3 | 4 | 5 | 6 | 7 | 8 | 9 |
|---|---|---|---|---|---|---|---|---|---|
| 1. Subjective Well-Being | - | .10** | .05** | .11** | .06** | .09** | -.02 | .18** | -.16** |
| 2. Prosocial Spending | | - | .17** | .02* | .21** | .05** | .03** | .12** | .13** |
| 3. Buying Time | | | - | .05** | .03** | -.16** | -.02 | .14** | .10** |
| 4. Resource Orientation[1] | | | | - | .06** | .02 | .01 | -.01 | -.07** |
| 5. Lay Theories[2] | | | | | - | .12** | .04** | -.03* | .20** |
| 6. Age | | | | | | - | -.09** | .19** | -.08** |
| 7. Gender | | | | | | | - | -.12** | .08** |
| 8. Personal Income | | | | | | | | - | -.20** |
| 9. Discretionary Income[3a5] | | | | | | | | | - |

* p < .01

** p < .001.

[1] 0 = prioritize money over time, 1 = prioritize time over money.

[2] 0 = happier buying something for themselves, 1 = happier buying something for others.

[3] 0 = Not living paycheck to paycheck, 1 = living paycheck to paycheck.

measure mood, the company asked participants to complete three items adapted from the Scale of Positive and Negative Experiences [SPANE; 26]. Participants were first asked to reflect on the past 24 hours and report "how often they felt positive, happy, or joyful" and "how often they felt negative, sad, or unpleasant" on a scale from 1 (*Almost Never*) to 5 (*Almost Always*). Next, participants rated their general mood in the past 24 hours using a sliding scale ranging from -50 (*a very unhappy emoji*) to 50 (*a very happy emoji*). Because the three items were measured in different units, we standardized each of the three items and averaged their z-scores to create an overall measure of mood ($\alpha$ = .88). In the validation data, this adapted measure of mood was strongly correlated with the 6-item measure of positive mood from the SPANE, $r$ = .69, $t(998)$ = 30.15, $p$ < .001. To measure life satisfaction, the company asked participants to rate their satisfaction with life using four items (e.g., "My life is pretty excellent right now, all things considered") adapted from the Satisfaction with Life Scale [SWLS; $\alpha$ = .89; 27]. The participants rated their agreement with each statement using a modified 5-point Likert scale where 1 = *No*; 2 = *Not Really*; 3 = *Meh*, 4 = *Mostly*; and 5 = *Yes*. In our validation data, this adapted measure of life satisfaction was strongly correlated with the original SWLS, $r$ = .70, $t(998)$ = 30.60, $p$ < .001. Finally, we created an overall measure of SWB by standardizing and averaging the z-scores for mood and life satisfaction. Consistent with past theorizing that mood and life satisfaction are related yet distinct components of SWB, the z-scores were moderately correlated with each other, $r$ = .55, $t(14487)$ = 79.85, $p$ < .001.

**Prosocial spending.** In previous large-scale survey research, prosocial spending has been measured using a single item from the Gallup World Poll [14]. To be consistent with this past work, the company assessed whether people spent money on others by asking participants whether or not they have donated to charity in the past month.

**Buying time.** To assess whether people spent money on time-saving purchases, the company asked participants to complete a measure taken directly from previous research [2]: "*In a typical month, do you spend any money on time-saving purchases? Specifically, do you spend any money with the primary intention of acquiring free time: a purchase that allows you to have more free time? For example, do you spend money to take a taxi instead of the bus, to purchase household services (e.g., lawn-mowing, laundry, or housecleaning services), to use online services (online accounting software and research services), or to purchase more expensive groceries from a closer grocery store?*"

**Resource orientation.** To assess whether participants prioritised time or money, the company asked participants to complete a well-validated measure of resource orientation [24]. In this measure, participants are introduced to one individual who values time more than money, and another individual who values money more than time, and then asked to indicate whose values more closely resemble their own.

**Lay theories about spending money on others.** Because there are no existing measures assessing lay theories about the benefits of spending money on others (vs. the self), the company created a face-valid item asking participants which type of purchase brings them more happiness: something they buy for themselves or something they buy for others.

**Demographics.** Participants reported basic demographic information including their age, gender, and personal annual income (before taxes) on a sliding range ranging from $0 to $150,000+. As an alternative measure of income, the company assessed participants' access to *discretionary* income by asking whether or not they were currently living from paycheck to paycheck.

## Results

### Does spending behavior predict SWB?

**Prosocial spending.** A majority of participants reported donating to charity in the past month (60.11%). Consistent with past research on the emotional benefits of spending money on others, participants who donated to charity in the past month reported higher levels of SWB (M = 0.08, SD = 0.83, n = 8701) compared to those did not donate to charity (M = -0.10, SD = 0.85, n = 6842), $b = 0.18$, 95% CI [0.15, 0.20], $t(15541) = 13.04$, $p < .001$, $d = 0.21$, 95% CI [0.18, 0.24]. This relationship held even after controlling for gender, age, and personal income, $b = 0.15$, 95% CI [0.12, 0.18], $t(13793) = 10.34$, $p < .001$.

**Buying time.** Slightly less than half of the participants reported making time-saving purchases in a typical month (47.11%). Consistent with past research on the emotional benefits of buying time, participants who spent money on time-saving purchases reported higher levels of SWB (M = 0.05, SD = 0.81, n = 6818) compared to those who did not (M = -0.04, SD = 0.86, n = 8725), $b = 0.08$, 95% CI [0.06, 0.11], $t(15541) = 6.02$, $p < .001$, $d = 0.10$, 95% CI [0.07, 0.13]. This relationship held even after controlling for gender, age, and personal income, $b = 0.06$, 95% CI [0.04, 0.09], $t(13793) = 4.45$, $p < .001$.

### Is the relationship between spending behavior and SWB moderated by demographic characteristics, and personal values and beliefs?

**Analytic strategy.** To test whether the relationship between spending behavior and SWB varied based on demographic characteristics and personal values and beliefs, we ran a series of linear regression models. We examined whether demographic variables, including age, gender, personal income, and whether participants lived from paycheck to paycheck, moderated the relationship between spending behavior and SWB. Each spending behavior was analyzed independently. Specifically, we tested (a) whether donating to charity interacted with each demographic variable, and (b) whether making time-saving purchases interacted with the same demographic variables, to predict SWB.

Taking a similar approach, we also examined whether personal values and beliefs, including whether participants believed that they would be happier spending money on others (vs. themselves) and whether they prioritized time versus money, moderated the relationship between each spending behavior and SWB. Specifically, we tested (a) whether donating to charity interacted with lay theories about spending money on others, and (b) whether making time-saving purchases interacted with prioritizing time versus money, to predict SWB.

**Table 2. The relationship between donating to charity and SWB by age group.**

| | Mean (SD) | | $b^*$ | $df$ | $t$ | $p$ | Cohen's $d^*$ |
|---|---|---|---|---|---|---|---|
| | No Donation | Donated | | | | | |
| Young Adults (18–35) | -0.12 (0.80) | 0.02 (0.80) | 0.14 [0.10, 0.17] | 7939 | 7.61 | < .001 | 0.17 [0.13, 0.22] |
| Middle-Aged Adults (36–55) | -0.14 (0.88) | 0.09 (0.84) | 0.24 [0.18, 0.29] | 4510 | 8.94 | < .001 | 0.27 [0.21, 0.33] |
| Older Adults (55+) | -0.009 (0.93) | 0.28 (0.87) | 0.29 [0.20, 0.38] | 1615 | 6.23 | < .001 | 0.32 [0.22, 0.43] |

$^*$95% CI in parentheses.

In light of our large sample size, main effects and interactions that fell above $p$ = .01 were treated as non-significant effects [for a brief explanation, see 28]. In addition, we did not include gender, age, and personal income as covariates in any of our models because the relationship between each spending behavior and SWB was virtually unchanged regardless of whether these variables were included in the model.

**Demographics and prosocial spending.** *Age.* The relationship between donating to charity and SWB varied with age, $b$ = 0.005, 95% CI [0.003, 0.007], $t(14066)$ = 4.42, $p$ < .001. Specifically, the relationship was stronger for older participants. For illustrative purposes, we looked at the relationship between donating to charity and SWB for young adults (18 to 35 years old; n = 7941), middle aged adults (36 to 55 years old; n = 4512), and older adults (more than 55 years old; n = 1617). Although the relationship was the strongest among older adults, the relationship was still significant within each group, $p$'s < .001 (see Table 2).

*Gender.* The relationship between donating to charity and SWB did not vary with gender, $b$ = -0.04, 95% CI [-0.11, 0.03], $t(14001)$ = -1.01, $p$ = .31. People who donated to charity reported higher levels of SWB, whether they identified as male ($b$ = 0.22, 95% CI [0.16, 0.28], $t(14001)$ = 6.95, $p$ < .001, $d$ = 0.26, 95% CI [0.19, 0.34]) or female ($b$ = 0.18, 95% CI [0.15, 0.21], $t(14001)$ = 11.33, $p$ < .001, $d$ = 0.22, 95% CI [0.18, 0.26]).

*Personal income.* Personal income did not moderate the relationship between donating to charity and SWB, $b$ < .001, $t(13859)$ = 0.47, $p$ = .64. This result held even after we applied a logarithmic transformation to personal income, $b$ = 0.02, 95% CI [-0.02, 0.06], $t(13667)$ = 1.03, $p$ = .30. In other words, participants who donated to charity consistently reported higher levels of SWB, regardless of income. To further understand these results, we examined the relationship between donating to charity and SWB for participants who made (a) between $0 to $28,000 (first quartile), (b) $28,001 to $41,000 (second quartile), (c) $41,001 to $60,000 (third quartile), (d) $60,001 to $150,000 (fourth quartile) per year, as well as (e) participants who fell below the poverty threshold of $13,300 for a single person under the age of 65 in 2019 [29]. Within each group, donating to charity was associated with higher levels of SWB—even among those who fell below the poverty threshold (see Table 3). In addition, the relationship between donating to charity and SWB was stronger for participants in the top (vs. bottom) income quartile, but this interaction failed to reach our threshold of significance, $b$ = 0.09, 95% CI [0.008, 0.17], $t(13855)$ = 2.16, $p$ = .03.

*Paycheck to paycheck.* In a similar vein, living paycheck to paycheck did not moderate the relationship between donating to charity and SWB, $b$ = -0.06, 95% CI [-0.15, 0.03], $t(14066)$ = -1.37, $p$ = .17. Among participants who did not live paycheck to paycheck, those who donated to charity in the past month reported higher levels of SWB (M = 0.59, SD = 0.73, n = 1097)

**Table 3. The Relationship between donating to charity and SWB by income.**

|  | Mean (SD) |  | $b^*$ | df | t | p | Cohen's $d^*$ |
|---|---|---|---|---|---|---|---|
|  | No Donation | Donated |  |  |  |  |  |
| $0 to $28,000 | -0.27 (0.86) | -0.15 (0.87) | 0.12 [0.06, 0.17] | 3562 | 3.95 | < .001 | 0.13 [0.07, 0.20] |
| $28,001 to $41,000 | -0.15 (0.82) | 0.02 (0.82) | 0.16 [0.10, 0.22] | 3466 | 5.67 | < .001 | 0.20 [0.13, 0.26] |
| $41,001 to $60,000 | -0.02 (0.81) | 0.13 (0.79) | 0.15 [0.10, 0.21] | 3541 | 5.49 | < .001 | 0.19 [0.12, 0.26] |
| $60,001 to $150,000 | 0.05 (0.81) | 0.26 (0.77) | 0.20 [0.15, 0.26] | 3286 | 6.96 | < .001 | 0.26 [0.19, 0.34] |
| < Poverty Threshold | -0.39 (0.86) | -0.22 (0.91) | 0.17 [0.06, 0.28] | 1009 | 3.00 | .003 | 0.19 [0.07, 0.31] |

*95% CI in parentheses.

compared to those who did not (M = 0.37, SD = 0.82, n = 501), $b$ = 0.22, 95% CI [0.14, 0.31], $t$ (14066) = 5.14, $p$ < .001, $d$ = 0.30, 95% CI [0.19, 0.40]. Notably, even participants living from paycheck to paycheck reported higher levels of SWB if they donated to charity in the past month (M = 0.002, SD = 0.81, n = 7379) than if they did not (M = -0.16, SD = 0.83, n = 5093), $b$ = 0.16, 95% CI [0.13, 0.19], $t$(14066) = 10.93, $p$ < .001, $d$ = 0.20, 95% CI [0.16, 0.23].

*Summary*. Taken together, we found limited evidence that the relationship between donating to charity and SWB was moderated by demographic characteristics. With the exception of age, donating to charity was associated with higher SWB regardless of gender, personal income, and whether or not participants lived from paycheck to paycheck. Furthermore, although the relationship between donating to charity and SWB was significantly stronger among older participants, donating to charity was associated with higher SWB within each age group (including young adults and middle-aged adults). Although our data are correlational, these results suggest that the happiness benefits of donating to charity do not hinge on certain demographic characteristics.

**Demographics and buying time.** *Age*. Age did not moderate the relationship between buying time and SWB, $b$ < .001, $t$(14066) = 0.55, $p$ = .59. Buying time was consistently associated with higher levels of SWB regardless of age (see Table 4).

*Gender*. The relationship between buying time and SWB did not vary with gender, $b$ = 0.02, 95% CI [-0.05, 0.09], $t$(14001) = 0.58, $p$ = .56. Looking within each gender, women who made time-saving purchases reported significantly higher levels of SWB (M = 0.04, SD = 0.81, n = 5202) compared those who did not (M = -0.05, SD = 0.86, n = 5949), $b$ = 0.09, 95% CI

**Table 4. The relationship between buying time and SWB by age group.**

|  | Mean (SD) |  | $b^*$ | df | t | p | Cohen's $d^*$ |
|---|---|---|---|---|---|---|---|
|  | Did not buy time | Bought time |  |  |  |  |  |
| Young Adults | -0.09 (0.81) | 0.01 (0.79) | 0.10 [0.07, 0.14] | 7939 | 5.61 | < .001 | 0.13 [0.08, 0.17] |
| Middle-Aged Adults | -0.05 (0.88) | 0.08 (0.84) | 0.13 [0.08, 0.18] | 4510 | 5.01 | < .001 | 0.15 [0.09, 0.21] |
| Older Adults | 0.15 (0.91) | 0.25 (0.86) | 0.10 [0.001, 0.19] | 1615 | 1.98 | .05 | 0.11 [0.001, 0.21] |

*95% CI in parentheses.

[0.06, 0.12], $t(14001) = 5.79$, $p < .001$, $d = 0.11$, 95% CI [0.07, 0.15]. Making time-saving purchases was also associated with higher levels of SWB within a smaller subsample of men ($M_{purchase} = 0.07$, $SD_{purchase} = 0.82$, $n_{purchase} = 1404$ vs. $M_{did\ not\ purchase} = -0.002$, $SD_{did\ not\ purchase} = 0.87$, $n_{did\ not\ purchase} = 1450$), but the relationship did not reach our threshold of significance, $b = 0.07$, 95% CI [0.01, 0.13], $t(14001) = 2.29$, $p = .02$, $d = 0.08$, 95% CI [0.01, 0.16].

*Personal income*. Personal income did not moderate the relationship between buying time and SWB, $b < -0.001$, $t(13859) = -1.21$, $p = .23$. This result held even after we applied a logarithmic transformation to personal income, $b = -0.003$, 95% CI [-0.04, 0.04], $t(13667) = -0.17$, $p = .87$. Looking within each income bracket, however, we did not find a significant relationship between buying time and SWB ($p$'s range: .02–.50; see Table 5). Given the relatively small relationship between buying time and SWB in the full sample, it is perhaps not surprising that the relationship failed to reach significance in smaller subsamples.

*Paycheck to paycheck*. Similarly, living paycheck to paycheck did not moderate the relationship between buying time and SWB, $b = 0.06$, 95% CI [-0.02, 0.15], $t(14066) = 1.40$, $p = .16$. To understand these results, we looked at the relationship between buying time and SWB for participants who lived from paycheck to paycheck and those who did not, respectively. Interestingly, participants who lived from paycheck to paycheck reported higher levels of SWB when they made time-saving purchases (M = -0.02, SD = 0.80, n = 5785) compared to those who did not make time-saving purchases (M = -0.10, SD = 0.84, n = 6687), $b = 0.08$, 95% CI [0.05, 0.11], $t(14066) = 5.22$, $p < .001$, $d = 0.09$, 95% CI [0.06, 0.13]. In contrast, we were unable to detect this difference in a smaller subsample of participants who did not live from paycheck to paycheck, $b = 0.02$, 95% CI [-0.06, 0.10], $t(14066) = 0.38$, $p = .70$.

*Summary*. Taken together, we found no evidence that the relationship between buying time and SWB was moderated by age, gender, and personal income. Most notably, making time-saving purchases was associated with higher levels of happiness—even for participants who lived from paycheck to paycheck.

**Personal beliefs and prosocial spending.** Does the relationship between donating to charity and SWB depend on whether people believe that they will be happier spending money on others (vs. themselves)? We did not find a significant interaction between donating to charity and lay theories about spending money on others, $b = 0.02$, 95% CI [-0.03, 0.08], $t(15539) = 0.81$, $p = .42$. People who thought spending money on others would lead to greater happiness reported higher levels of SWB if they donated to charity in the past month (M = 0.10, SD = 0.83, n = 6525) than if they did not (M = -0.07, SD = 0.85, n = 3753), $b = 0.17$, 95% CI

**Table 5. The relationship between buying time and SWB by income.**

| | Mean (SD) | | | | | | |
|---|---|---|---|---|---|---|---|
| | No Donation | Donated | $b^*$ | df | t | p | Cohen's $d^*$ |
| $0 to $28,000 | -0.23 (0.88) | -0.17 (0.85) | 0.06 [-0.002, 0.11] | 3562 | 1.89 | .06 | 0.06 [0.002, 0.13] |
| $28,001 to $41,000 | -0.06 (0.83) | -0.04 (0.81) | 0.02 [-0.04, 0.07] | 3466 | 0.68 | .50 | 0.02 [0.04, 0.09] |
| $41,001 to $60,000 | 0.04 (0.83) | 0.10 (0.78) | 0.06 [0.009, 0.11] | 3541 | 2.30 | .02 | 0.08 [0.01, 0.14] |
| $60,001 to $150,000 | 0.16 (0.83) | 0.22 (0.75) | 0.05 [-0.002, 0.11] | 3286 | 1.90 | .06 | 0.07 [0.002, 0.14] |
| < Poverty Threshold | -0.34 (0.92) | -0.24 (0.85) | 0.10 [-0.01, 0.21] | 1009 | 1.75 | .08 | 0.11 [0.01, 0.24] |

$^*$95% CI in parentheses.

[0.14, 0.20], $t(15539)$ = 9.90, $p < .001$, $d$ = 0.20, 95% CI [0.16, 0.24]. Remarkably, even participants who thought spending money on themselves would lead to greater happiness reported higher levels of SWB if they donated to charity (M = 0.01, SD = 0.80, n = 2176) than if they did not (M = -0.13, SD = 0.85, n = 3089), $b$ = 0.15, 95% CI [0.10, 0.19], $t(15539)$ = 6.24, $p < .001$, $d$ = 0.18, 95% CI [0.12, 0.23]. In other words, donating to charity was associated with higher levels of SWB—whether participants believed that they would be happier spending money on other or not. Thus, donating to charity may be a particularly powerful strategy for increasing SWB.

**Personal values and buying time.** Does the relationship between buying time and SWB depend on whether people value time versus money? There was a significant interaction between making time-saving purchases and prioritizing time (vs. money)—but in a counterintuitive direction, $b$ = -0.07, 95% CI [-0.13, -0.02], $t(14056)$ = -2.51, $p$ = .01. Specifically, we found that buying time was more strongly associated with higher levels of SWB among participants who valued money over time; among individuals who prioritized money, those who made time-saving purchases reported higher levels of SWB (M = -0.02, SD = 0.82, n = 3248) compared to those who did not (M = -0.14, SD = 0.86, n = 3983), $b$ = 0.11, 95% CI [0.07, 0.15], $t(14056)$ = 5.76, $p < .001$, $d$ = 0.13, 95% CI [0.09, 0.18]. Participants who valued time also reported higher levels of SWB when they made time-saving purchases (M = 0.12, SD = 0.80, n = 3395) than when they did not (M = 0.07, SD = 0.84, n = 3434), but this relationship failed to reach our threshold of significance, $b$ = 0.04, 95% CI [0.003, 0.08], $t(14056)$ = 2.12, $p$ = .03, $d$ = 0.05, 95% CI [0.005, 0.10]. This tentative finding points to the interesting possibility that buying time may be especially effective for increasing SWB among those who overlook the benefits of having more time relative to having more money.

## General discussion

By drawing on a large and economically diverse sample of Americans, the present research provides some of the clearest evidence to date that prosocial spending and buying time are robustly related to happiness within the United States. Individuals who donated money to charity were happier than those who did not, and this relationship was significant for both men and women, and across the income spectrum, including for participants who reported living from paycheck to paycheck. Indeed, even participants who fell below the poverty threshold reported greater happiness if they donated money to charity than if they did not. The relationship between prosocial spending and happiness was moderated by age, with older adults exhibiting a stronger positive effect of donating to charity compared to younger adults; this finding dovetails with previous theorizing suggesting that prosocial behavior should be especially rewarding in old age [e.g., 21]. Even among young adults though, donating to charity was linked to significantly greater happiness. Indeed, the relationship between prosocial spending and happiness was highly significant ($p < .001$) within every demographic group we examined.

By contrast, the relationship between buying time and happiness was less reliable within some demographic groups. Although the relationship between buying time and happiness was not moderated by gender or income, the link between buying time and happiness fell just short of significance for men and failed to reach significance within each income bracket. Interestingly though, the relationship between buying time and happiness held for individuals who were living paycheck-to-paycheck, suggesting that time-saving purchases may be beneficial even for individuals who are struggling to make ends meet. The effect of buying time on happiness did not depend on age, and the positive relationship between buying time and happiness was highly significant for every age group. Taken together, our findings suggest that the positive relationship between buying time and happiness is fairly reliable for different

demographic groups, while the relationship between donating to charity and happiness is highly reliable.

Going beyond demographics, we also examined the moderating role of individuals' personal values and beliefs. Remarkably, we found that the effects of donating to charity were not moderated by individuals' personal beliefs; even people who believed they would be happier spending money on themselves (vs. others) were happier if they donated money to charity than if they did not. This finding provides tentative evidence that even those who think they are better off spending money on themselves may still benefit from spending on others. We also asked participants whether they placed greater value on time or money, and responses to this question did moderate the effect of buying time on happiness. Interestingly, people who prioritized money over time were happier if they used money to buy time than if they did not. This somewhat unintuitive finding suggests that people who chronically value money over time might benefit from the recommendation to give up some of their money in order to have more time. Meanwhile, individuals who valued time over money were equally happy whether or not they used money to buy time. We would speculate that people who chronically prioritize time may make a variety of beneficial choices (such as opting to work fewer hours), thereby limiting the additional impact of making time-saving purchases. To test this possibility, it would be worthwhile to conduct experimental research, recruiting participants who chronically value time versus money, and randomly assigning them to buy time (or not).

More broadly, experimental research is essential for confirming whether prosocial spending and buying time exert a consistent causal impact on happiness for the different segments of the population examined here. Because the present research was correlational and our analyses were not pre-registered, all of our findings should be treated as exploratory and generative. Still, considering the very high cost of conducting spending experiments (in which participants are typically given money to spend) and of properly powering these experiments to detect interaction effects (which typically requires thousands of participants), large-scale correlational research provides an important foundation. Moreover, correlational research in this area is valuable because this approach enables us to examine the relationship between happiness and people's own spending decisions in daily life using their own hard-earned money (rather than windfalls provided by a researcher).

This naturalistic approach enables us to establish the effect sizes for the relationship between happiness and our spending strategies, which should be fairly stable and precise estimates given our large sample size. The effect of prosocial spending on happiness was small ($d = 0.21$), but approximately twice the size of the effect of buying time on happiness ($d = 0.10$), which was very small. As Funder and Ozer [30] have argued however, even seemingly diminutive effects can exert an important influence over the long-term, especially if these patterns of behavior are repeated. Given the numerous spending decisions presented by everyday life, these small effects have the potential to play a non-trivial role in shaping human happiness. That said, happiness is shaped by numerous factors, from genetics [for a review, see 31] and employment status [for a review, see 32] to brief social interactions [e.g., 33,34], and thus we would be skeptical if the effect of spending decisions on happiness appeared to be very large. Indeed, the effect of prosocial spending (versus buying time) may be somewhat inflated because prosocial spending may reflect a broader other-oriented approach to life, whereas buying time represents a narrower spending choice. It is also worth noting that our effects may have been slightly attenuated due to the use of adjusted well-being measures (which were "on brand" for the company conducting the survey), although the fact that these measures were highly correlated with well-validated measures is reassuring.

Finally, it is essential to note that our research only included Americans, but the present work provides a valuable comparison point for studies conducted in other cultural contexts.

Our research also did not enable us to examine the full range of demographic variables, and future work should examine the role of race/ethnicity and cultural background, among other dimensions. Still, our research provides some initial reassurance that popular recommendations to spend on others and buy time may be reliably linked to happiness across diverse segments of American society, including people with little discretionary income.

## Author Contributions

**Conceptualization:** Iris Lok, Elizabeth W. Dunn.

**Data curation:** Iris Lok.

**Formal analysis:** Iris Lok.

**Supervision:** Elizabeth W. Dunn.

**Writing – original draft:** Iris Lok, Elizabeth W. Dunn.

**Writing – review & editing:** Iris Lok, Elizabeth W. Dunn.

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
