## [Decision Letter · Decision Letter 0]

10 Mar 2022

PONE-D-22-04151Are the Benefits of Prosocial Spending and Buying Time Moderated by Age, Gender, or Income?PLOS ONE

Dear Dr. Lok,

Thank you for submitting your manuscript to PLOS ONE. After careful consideration, we feel that it has merit but does not fully meet PLOS ONE’s publication criteria as it currently stands. Therefore, we invite you to submit a revised version of the manuscript that addresses the points raised during the review process. Two experts in the field reviewed your submission and I also, independently, read it. On the whole, there is very strong support for seeing this work published in PLOS ONE. As you can read for yourself, the reviewers were generally very positive and their comments fell under, what I consider, a minor revision. I direct you to the reviews themselves for more details, but the points that I pulled out and concur with is that there needs to be more precision in your language (R1 + R2), the claims needs to be toned down a bit more and positioned better given existing literature (R1), the title likely needs a bit of repositioning (R2), there could be a bit more in terms of data/analysis reporting (R1), and the sample composition could be both better specified in relation to previous work, and more accurately described (it's not all that "diverse") (R2).  These are all changes that I have confidence the authors can handle, and, short of something major popping up in a revision, I do not anticipate sending the paper out to the reviewers again.

We look forward to receiving your revised manuscript.

Kind regards,

Jeff Galak, PhD

Academic Editor

PLOS ONE

Journal Requirements:

3. Please amend your current ethics statement to address the following concerns:

a) Did participants provide their written or verbal informed consent to participate in this study?

4. Please update your submission to use the PLOS LaTeX template. The template and more information on our requirements for LaTeX submissions can be found at http://journals.plos.org/plosone/s/latex.

Reviewers' comments:

Reviewer's Responses to Questions

**Comments to the Author**

1. Is the manuscript technically sound, and do the data support the conclusions?

Reviewer #1: Yes

Reviewer #2: Yes

2. Has the statistical analysis been performed appropriately and rigorously? 

Reviewer #1: Yes

Reviewer #2: Yes

3. Have the authors made all data underlying the findings in their manuscript fully available?

Reviewer #1: Yes

Reviewer #2: Yes

4. Is the manuscript presented in an intelligible fashion and written in standard English?

Reviewer #1: Yes

Reviewer #2: Yes

5. Review Comments to the Author

Reviewer #1: Are the Benefits of Prosocial Spending and Buying Time Moderated by Age, Gender, or Income?

This paper tests for moderation (by age, gender, income) of the effects of donations and buying time on subjective well-being. I have the following comments:

My most substantive comment is the overall SWB measure. It’s composed of a mood measure and a life satisfaction measure, but mood is measured over the past 24 hours, while life satisfaction is more open-ended. Why, and is there precedent for this? Throughout the paper, there are also references to “emotional benefits” – is this supposed to be the same thing as SWB? Some more precision seems desirable.

I realize the authors made up the item, but asking participants what type of purchase brings them more happiness: something they buy for themselves or something they buy for others is a lay theory more than anything. And the work of Gilovich and Epley (for example) show these beliefs are often misguided. I realize the authors are not using it as a measure of actual happiness or SWB, but they also call it a “preference” which I’m not sure it is either (participants are not asked where they would prefer to spend, but rather than they think would produce more happiness if they did).

Some other tweaking of the positioning may be necessary. For example, the authors write “However, it is unclear whether these strategies should be widely recommended to diverse demographic groups”, but the paper here looks at a single country, while other studies (e.g., Aknin et al 2013 JPSP, 2015 JEPG) have much more diverse demographics than what is here. I guess I am not sold on the claim that “the present research provides perhaps the clearest evidence to date that prosocial spending and buying time are robustly related to happiness across demographic groups”, when, at least for spending money on happiness, there are more heterogeneous samples reported.

Is effect of income on SWB actually stronger than effect of making a time saving purchase? Please report the correlation between time saving purchases and income, which should be related (more wealthy people outsource “help”).

Please clarify what here is predicted versus exploratory. There are some statements suggesting predictions, e.g., “but in an unexpected direction”, “because older adults are more likely to be retired, they might have more free time—and feel more time affluent—compared to their younger counterparts”.

The first sentence of abstract is a bit out of place, given the authors cite a plethora of past research showing the affirmative, and the specific question posed isn’t the focus of the present work.

Best wishes.

Reviewer #2: This manuscript examines whether the well-being benefits that come from prosocial spending and buying time are moderated by individual demographic factors, such as age, gender, and income. The paper is clear, on an important topic, and well-written. The authors also utilize an impressively large sample size and deserve praise for their efforts. Although the evidence presented is only correlational in nature (which these researchers are transparent about; to their credit, they are refreshingly careful not to overclaim or make too much of their empirical results), I do think these data would be of interest to other scholars who work in this area of inquiry. Indeed, on page 21, the authors themselves write, “all of our findings should be treated as exploratory and generative.” I agree. Given that this particular outlet places less emphasis on evaluations of “impact” than other journals, I will focus my comments on just a few ways this contribution can be improved.

First, the authors include citations to the literature on buying experiences to motivate their study. This description of how understanding evolved in this space is a bit misleading. In their initial 2003 JPSP, Van Boven and Gilovich actually initially looked at income effects, and that very first publication on this idea found that the “experiential advantage” was not present among those at the very lowest end of the income distribution. As a result, it has long been suggested that such effects are really about money spent with one’s discretionary income. Notably, those results were also based on a representative sample of American respondents rather than the convenience sample used in the current research. This distinction is relevant to the present work given that nearly 80% of the sample recruited was female [Van Boven & Gilovich (2003) also investigated potential gender effects, and did not observe differences between genders], and that nearly 90% of the sample here reported having little to no discretionary income.

In addition, on multiple occasions, these researchers refer to their sample as “diverse.” It could be worth mentioning what they mean when they use the word “diverse” here. Nothing is said about culture, race, ethnicity, political orientation, geographic location, and so on. Of course, the writing does briefly note that these factors were not explored, but more could be included on this.

Finally—and as a minor suggestion—I think it would be wise to have a more informative title that provides some answers in the form of a declarative statement rather than one which asks a question. The current title reads a bit more like the headline of an article in the popular press (such as those in cited references 3-8) instead of an academic paper. I thank the authors for their submission and wish them luck as they continue to work on their interesting research.

6. PLOS authors have the option to publish the peer review history of their article (what does this mean?). If published, this will include your full peer review and any attached files.

Reviewer #1: No

Reviewer #2: No

---

## [Author Response · Author response to Decision Letter 0]

28 Apr 2022

Please see attached rebuttal letter (labelled "Response to Reviewers").

---

## [Editor Report · Decision Letter 1]

25 May 2022

Are the Benefits of Prosocial Spending and Buying Time Moderated by Age, Gender, or Income?

PONE-D-22-04151R1

Dear Dr. Lok,

We’re pleased to inform you that your manuscript has been judged scientifically suitable for publication and will be formally accepted for publication once it meets all outstanding technical requirements.

Kind regards,

Jeff Galak, PhD

Academic Editor

PLOS ONE
---

## [Editor Report · Acceptance letter]

2 Jun 2022

PONE-D-22-04151R1 

Are the Benefits of Prosocial Spending and Buying Time Moderated by Age, Gender, or Income? 

Dear Dr. Lok:

I'm pleased to inform you that your manuscript has been deemed suitable for publication in PLOS ONE. Congratulations! Your manuscript is now with our production department. 

Kind regards, 

on behalf of

Dr. Jeff Galak 

Academic Editor

PLOS ONE